# Comparison of Electrodermal Activity from Multiple Body Locations Based on Standard EDA Indices’ Quality and Robustness against Motion Artifact

**DOI:** 10.3390/s22093177

**Published:** 2022-04-21

**Authors:** Md-Billal Hossain, Youngsun Kong, Hugo F. Posada-Quintero, Ki H. Chon

**Affiliations:** Department of Biomedical Engineering, University of Connecticut, Storrs, CT 06269, USA; md.b.hossain@uconn.edu (M.-B.H.); youngsun.kong@uconn.edu (Y.K.); hugo.posada-quintero@uconn.edu (H.F.P.-Q.)

**Keywords:** electrodermal activity, SCR, statistical test, cognitive stress

## Abstract

The most traditional sites for electrodermal activity (EDA) data collection, palmar locations such as fingers or palms, are not usually recommended for ambulatory monitoring given that subjects have to use their hands regularly during their daily activities, and therefore, alternative sites are often sought for EDA data collection. In this study, we collected EDA signals (*n* = 23 subjects, 19 male) from four measurement sites (forehead, back of neck, finger, and inner edge of foot) during cognitive stress and induction of mild motion artifacts by walking and one-handed weightlifting. Furthermore, we computed several EDA indices from the EDA signals obtained from different sites and evaluated their efficiency to classify cognitive stress from the baseline state. We found a high within-subject correlation between the EDA signals obtained from the finger and the feet. Consistently high correlation was also found between the finger and the foot EDA in both the phasic and tonic components. Statistically significant differences were obtained between the baseline and cognitive stress stage only for the EDA indices computed from the finger and the foot EDA. Moreover, the receiver operating characteristic curve for cognitive stress detection showed a higher area-under-the-curve for the EDA indices computed from the finger and foot EDA. We also evaluated the robustness of the different body sites against motion artifacts and found that the foot EDA location was the best alternative to other sites.

## 1. Introduction

In recent years, there has been a tremendous increase in the popularity of wearable devices, which has greatly increased the feasibility of non-invasive and continuous physiological data collection [1,2]. Electrodermal activity (EDA) is one example, which has been used as a non-invasive surrogate marker of the autonomous nervous system in several psychophysiological applications, such as emotional arousal [3,4,5,6], stress [7,8,9,10], pain [11,12,13], panic disorder [14], autism [15], and decision making [16]. EDA refers to the change in electrical conductivity of the skin in response to eccrine sweat gland activity. Usually, EDA is collected by applying a low constant voltage between two closely placed electrodes and then measuring the change in skin conductance [7,17,18]. Typically, EDA electrodes are placed on the fingers of the subject. However, fingers may not be practical locations for long-term data collection. Therefore, alternative body sites should be explored for EDA data collection when fingers are not available. The objective of this study is to determine alternative sites for EDA collection other than fingers by quantitative comparison of the EDA indices derived from other body locations. In addition, the other aim is to determine the effect of common motion artifacts on the fidelity of EDA data for various body locations examined.

The most widely used and the most responsive sites for EDA signal measurement are the finger pads and palmar surfaces [17,18,19]. However, neither are feasible for applications in which subjects have to use their hands. For example, a recent study has shown promising results on the prediction of seizures due to central nervous system oxygen toxicity in rats using indices derived from EDA [20]. Given that divers use their hands regularly for performing many different tasks, the palmar surface is not the best site for collecting EDA signals. Moreover, there could be some conditions for which the finger EDA may not provide the best response. For example, Sano et al. [21] reported that while monitoring non-REM sleep storm activity, they found that larger and more frequent skin conductance responses were observed in the EDA records collected from wrists than from the palm.

Skin conductance at a particular body site depends on several factors, such as the density of sweat glands and their relative sizes, the sweat output of the individual glands, and also the nerves that control the sweat glands of the site [22,23]. Palmar and plantar surfaces have the highest eccrine sweat gland density (600 to 700 glands/cm^2^) and are often considered the best sites for EDA recordings [24,25,26]. Another high sweat gland density site is the forehead (eccrine gland density of 181 glands/cm^2^), which has also been explored by researchers for EDA data collection [22,26].

As an alternate measurement site, the wrists were suggested by many researchers in the past few years [21,27,28]. However, contradictory results were reported by the researchers for this measurement site. While some of the studies showed high correlations between the EDA signals collected from the wrist and the palmer sites [22,28], others found lower correlations [29,30]. Researchers have also considered several other sites, such as shoulders [22,31] and the lower calf [22,31,32], for EDA data collection. For example, Hedman et al. [33] collected EDA data from the lower calf of children with attention deficit hyperactivity disorder and reported that this particular location was beneficial for them since it did not interfere with their activities and movements. Fedor et al. [34] compared EDA signals from a forearm and the back of the lower calves and found significant correlations. The participants in this study rated the lower calf as more comfortable than the distal forearm location. However, several other studies found relatively low correlations between the EDA signal obtained from the lower calf and the gold standard finger EDA signals [22,32].

Another important factor to be considered about the EDA for finding alternate sites is the hydration time [32]. Depending on the location, a hydration time of 25 to 120 min might be necessary for obtaining accurate skin conductance responses (SCR) [29,32]. For example, Kasos et al. [32] reported that pedaling on a stationary ergometer for approximately 20 min has improved the absolute response frequency of the left shoulder to 96%. All of these studies suggest that for the alternative sites to be electrodermally active, enough hydration time or physical activity is needed.

While previous studies [22,32] have explored the feasibility of several alternate sites for EDA signal collection, their analysis was mostly based on morphological correlations with the gold standard finger EDA. They did not compare the EDA indices calculated from different sites. While correlation with the finger EDA is a practical way of evaluating the EDA signals at alternative sites, it can be misleading sometimes because of the trend of both signals. For example, EDA signals having similar trends (increasing or decreasing) may show high correlation even without similar SCR responses. Moreover, many of the studies did not consider the effect of motion artifacts and how they can negatively affect the EDA records obtained from different body sites.

In this study, we considered four different body sites (forehead, back of neck, finger, and foot) that have high sweat gland density, and these sites were also previously reported to have highly correlated EDA signals when compared to the finger EDA [22,30,32,35]. While comparing the signals obtained from alternative sites to the finger EDA, we also analyzed the usability of the EDA signals under daily activities such as walking or grabbing something. We computed EDA indices and compared their efficacy at classifying cognitive stress for all data collected from the four different body sites.

## 2. Materials and Methods

### 2.1. Data Collection

We collected four channels of EDA signals from different body locations, namely the forehead, neck, fingers, and foot. Figure 1 shows the measurement sites for this study. In the study, 23 subjects aged 20–35 (19 male, 4 female) participated. We used the non-dominant side for placing electrodes on the index and middle fingers and the inner foot.

Similar to most studies [22,30,32,35], we collected EDA data using an exosomatic approach, where an external constant voltage or current is applied between two electrodes. A constant DC/AC voltage/current is applied between the electrodes, and the corresponding current/voltage is measured, from which skin conductance (SC) is calculated using the Ohm’s law. The variation of the SC represents the EDA signal. We used commercially available wearable Shimmer 3 devices [36] with Ag/AgCl electrodes. The Shimmer 3 device uses a constant small DC voltage across the electrodes, measures the corresponding current, and computes the SC. The SC is amplified to provide EDA data within a standard range. This device is easy to handle and worn as a wearable device [37]. The experimental protocols were described to the subjects by the experimenter and written consent was obtained from the subjects before starting the experiment. The experimental protocols were reviewed and approved by the institutional review board (IRB) for human subject research at the University of Connecticut.

The experimental protocols for this study are shown in Table 1. The experimental protocols consisted of two parts. Part I was designed to compare the EDA collected under no movement as subjects were resting in the supine position for two minutes, and then subjects performed the Stroop test in the same position (supine position). The experimental phases were performed one after another with no pause in between. For each phase, we considered 2 min, which is sufficiently enough to contain many SCRs. The Stroop color and word test (SCWT) is a neurophysiological test to assess a person’s ability to handle cognitive stimuli [38]. We used the most common version of SCWT. Subjects were shown various color-words printed in an inconsistent color ink (for instance, the word “red” is printed in green ink) and were required to name the color of the ink instead of reading the word. We designed part II of the protocol to evaluate the signal quality during movements that emulated realistic scenarios: we considered two basic movements such as walking and lifting a weight in their hand. While the movements we considered may seem limited, our main focus was on how the EDA signal can be affected at the fingers and the foot due to common movements. Since walking is one of the most common activities that may affect the EDA signal at the foot, and we often use our hands for holding or lifting objects that may affect the EDA signals at the fingers, we considered these two movements to particularly explore the signals at the fingers and the foot during movements or regular activities.

### 2.2. Data Processing

The EDA data collected were resampled at 8 Hz from 100 Hz. We then used a Butterworth lowpass filter of order six and cutoff frequency of 0.6 Hz (which is well above the frequency range of EDA dynamics) [39]. Since typically the autonomous nervous activities are captured within the frequency range (<0.4 Hz) [40,41], the choice of a cutoff frequency of 0.6 Hz was a relatively conservative approach to remove redundancy in the data while keeping the essential information. We decomposed the EDA signal into phasic and tonic components using the cvxEDA algorithm [42], which is one of the popular methods for EDA signal decomposition. The cvxEDA models the EDA signal as a summation of phasic and tonic components, and an additive white Gaussian noise term which incorporates the prediction as well as the measurement error. This method then uses Bayesian statistics, convex optimization, and sparsity constraint to solve the optimization problem to minimize the prediction errors to estimate the tonic and phasic components. Phasic and tonic components are the two most salient characteristics of an EDA signal, where the phasic components (SCRs) represent the rapid and smooth transient events present in the EDA signals, and the tonic component represents the overall conductance as a measure related to the slow shifts of the EDA [43].

We then used the comments noted during the experiment to segment the data into different stages, where each segment corresponds to a particular phase of the experiment (e.g., baseline, Stroop test, walking, and weightlifting). We computed traditional EDA indices, such as the number of skin conductance responses, mean and variance of skin conductance level, and the mean and variance of phasic signals, from each segment of the four EDA locations [43]. Since the EDA signals collected from different locations may have different signal amplitudes while containing the SCRs, we hence used 0.05 of the maximum peak amplitude of the respective phasic signal as the threshold to compute the number of skin conductance responses.

### 2.3. Statistical Analysis

Many of the previous publications reported Pearson’s correlation to compare the EDA signals from different body sites. However, as mentioned in [32], correlations may overestimate or underestimate the relationship between two measurement sites, depending on the trends of the signals. While the trends may represent important information about the skin conductance, such as how the overall activity changes over time, they can be misleading in certain cases. For example, Figure 2 left shows a segment of the four simultaneous EDA records collected from different body sites. While from the raw EDA signal it seems that the neck and forehead EDA follow the same trend as the finger EDA for the first half of the time series, the phasic components derived from them look quite different than those of the finger EDA for all time points.

The phasic signal, which is believed to contain most of the information, does not always match very well to the corresponding finger phasic signal (especially in the first half). Therefore, for a more comprehensive comparison between the measurement sites, we computed correlations for the phasic and tonic components among the signals from different measurement sites.

While within-subject correlation has been reported in most of the previous studies, none of them compared EDA-derived indices that are typically used for the identification of sympathetic stimuli. We computed EDA indices from each body location’s signals and compared their efficacy at differentiating between the baseline stage and the cognitive stress stage (SCWT). To check the statistical difference of the EDA indices between the baseline and the cognitive stress stages, we performed a pairwise t-test if the data of two groups were normally distributed. Otherwise, we used the two-sided Wilcoxon rank sum test [44]. We used a 0.05 level of significance for rejecting the null hypothesis. A significant difference between the two groups indicated the sensitivity of the EDA from a particular body site to an external stimulus.

### 2.4. EDA Data Quality Assessment

One of the significant motivations behind exploring alternative sites for EDA data collection is to find a comfortable body site for electrode placement that is also robust against motion artifacts. Since a major focus is on ambulatory EDA monitoring, it is essential to analyze the effect of daily movements on the EDA collected from different measurement sites [45,46]. In this study, we considered two common motion artifact scenarios (walking and a grabbing motion using hands) for EDA data collection.

To compare the data quality among different body locations, we first computed the power spectrum of the EDA signal using the Welch periodogram with 50% data overlap, and then defined a noise band (frequency > 0.4 Hz). The reason we choose 0.4 Hz is that most of the ANS activity (sympathetic and parasympathetic) is captured within the frequency range of 0.04 to 0.4 Hz [39,40]. Therefore, we can attribute the spectral power in the frequency > 0.4 Hz to either noise or dynamics not related to the EDA signal. Thus, we defined a metric $P_{n}$, which is the fraction of the total power present in the noise band:(1)$$P_{n}=\frac{\mathrm{Power}\;\mathrm{in}\;\mathrm{noise}\;\mathrm{band}\left(f>0.4\right)}{\mathrm{Total}\;\mathrm{power}}\;$$

The higher the $P_{n}$, the higher the motion artifacts or noise that is present in the signal. We compared the EDA signals obtained from different body sites in terms of $P_{n}$.

## 3. Results

### 3.1. Correlation with the Finger EDA

We computed the Pearson correlation between the finger EDA and EDAs from other body sites during the “no movement” phases (i.e., baseline and SCWT). Since during the first two phases of the protocol there is no significant motion artifact, for accurate correlation we used continuous 4 min data (baseline and SCWT) for calculating the Pearson correlation. Table 2 shows the within-subject correlations (correlation is computed among the EDA channels of the same subject) between the finger EDA and the EDAs from the alternate sites. Similar to most previously reported articles, we also obtained a high correlation between EDA signals from the finger and the foot. However, unlike many other published reports, we did not observe a high correlation of the neck and forehead EDA with the fingers.

We also observed a larger between-subjects variability (higher variability across the subjects: evident from the higher standard deviation value), which suggests that even though for some subjects the forehead and neck EDA were highly correlated with the finger EDA, for others this was not the case. As mentioned earlier, the correlation using only the raw data could be misleading because of the underlying trends in the data; hence, we also computed correlation for the phasic and the tonic components of the EDA. As shown in Table 2, compared to other locations, feet EDA also showed a higher correlation with finger EDA when the phasic and tonic components from these two sites were compared separately. On the other hand, the neck and forehead again showed low correlation with the finger EDA for both tonic and phasic components. It can also be seen that the tonic components have slightly higher correlation (with high variability, though) when compared to phasic components for all body locations. This suggests that EDA at the forehead and neck are less responsive compared to EDA obtained from either the finger or the foot.

### 3.2. Separation of Cognitive Stress from the Baseline

We evaluated the efficacy of the EDA calculated from each measurement site to determine its responsiveness to cognitive stress. In this experiment, cognitive stress was induced by the SCWT, which can be considered a mild stress. We first performed the one-sample Kolmogorov–Smirnov test [47] on the data, and if normality was found, we performed the statistical t-test between baseline and the SCWT stage for each EDA index. Otherwise, we used the Wilcoxon rank sum test [44]. All the statistical tests were performed across the subjects over 23 samples. We used the 0.05 level of significance for rejecting the null hypothesis. Table 3 shows the summary results for each measurement site.

It can be seen from Table 3 that none of these EDA metrics showed significant differences between the baseline and the SCWT for the forehead and neck EDA. In the case of finger EDA, the number of SCR, mean, and standard deviation of the phasic signal showed significant differences between the baseline and SCWT. Four of the five metrics showed significant differences in the foot EDA signal.

We also evaluated EDA metrics for classifying differences between the baseline and the SCWT stages. We computed the receiver operating characteristic (ROC) curves for EDA metrics to classify between the baseline and the cognitive test stages. We first computed the true positive rate and the false positive rate for each feature at a variety of thresholds. The ROC curve was then obtained by plotting the true positive rate against the false positive rate. In this case, the positive class refers to the cognitive stress stage and the negative class represents the baseline stage. Figure 3 shows ROC curves for the number of SCR and the mean of the phasic signal. As expected, the finger and foot EDA exhibited the top two lines in both figures, with higher areas-under-the-curves than those of the forehead and neck EDAs.

### 3.3. Effect of Motion Artifacts

In order to evaluate the effect of motion artifacts on EDA signals from different measurement sites, we computed the power spectrum of the EDA signals during motion artifact and the fraction of total power in the noise band (frequency > 0.4), $P_{n}$, as defined earlier. As during the first two stages (baseline and SCWT) there was no significant motion, we only compared the $P_{n}$ during walking and weightlifting stages.

Figure 4 shows the comparison of $P_{n}$ values among the four different measurement sites. As can be seen from Figure 4 left, during walking, the EDA signals obtained from the foot had the highest power in the noise band, which is expected, since the foot is subjected to higher movement compared to other locations during walking. However, the motion artifacts in the foot EDA during walking are largely high-frequency noise that can be easily filtered out (as shown in Figure 5), which is why there was a high correlation (correlation between finger and foot raw EDA, tonic, and phasic components were found to be 0.6517, 0.6758, and 0.6971, respectively) between foot and finger EDA after lowpass filtering.

During weightlifting, the neck EDA was affected the most by motion artifacts, which could be because of the periodic contraction and expansion of the muscles in the neck. Since the subjects used their hands for lifting weights, the finger EDAs were distorted on many occasions, as expected. Figure 5 shows an example of the finger EDA during weightlifting, where it is evident that the finger EDA was highly distorted because of the movement of electrodes while holding and lifting the weight. We only showed the finger and foot EDA in Figure 5 since they are the only locations that were highly correlated.

As can be seen from Figure 4, the forehead is the most robust site against motion artifacts, as evidenced by the relatively low spectral power in the noise band compared to other measurement sites. The neck, on the other hand, was found to be highly sensitive to motion artifacts.

## 4. Discussion

In this study, we collected EDA signals from four measurement sites (namely forehead, neck, finger, and the foot) that are known to have high eccrine gland density. This study is one of the first studies to compare popular EDA indices obtained from different measurement sites. We also compared the robustness of different body sites against measurement noise.

Unlike many studies that only computed correlations between raw EDA signals, we computed within-subject correlations for the raw EDA and tonic and phasic components, which eliminated the risk of false correlations due to the trends in the signal. The within-subject correlation was the highest between the finger and the foot. The forehead and neck EDA showed moderate correlation with the finger EDA. However, for the phasic component, the correlation was rather low. This could be due to a lack of sufficient hydration time, which is required for the sweat glands to be innervated.

The effect of hydration time on the forehead EDA can be seen when we examine the number of subjects that showed a correlation higher than 0.5 between the finger and the forehead or the neck in different stages starting from the baseline. Figure 6 shows the progression of the number of subjects showing a correlation greater than 0.5 in different stages. As shown, as time progressed, more subjects had higher correlations between the forehead and finger EDA, which indicates that with sufficient hydration, the forehead may become more responsive and provide more accurate SCRs. It should be noted that, during weightlifting, the finger EDA was distorted on several occasions, which is why we did not compute the correlation during weightlifting.

We computed several key EDA indices from different body sites and compared their efficacy to differentiate between the baseline and cognitive stress (in this case, SCWT) stages. We observed that none of the EDA indices showed significant differences between the baseline and cognitive stress for the forehead and neck EDA. On the other hand, EDA indices computed from the finger and foot showed significant differences between the baseline and SCWT stages. The ROC curve computed using the EDA indices from the finger and the foot had a higher area-under-the-curve. SCWT induces mild cognitive stress and for some subjects, they may not experience any stress, which explains why the EDA from the finger and the foot showed moderate accuracy to differentiate stress from the baseline condition.

The effect of motion artifacts on the EDA signals obtained from different measurement sites shows that the neck is the most sensitive to motion artifacts, while the forehead is the most robust. However, the forehead needs a prolonged amount of hydration time for producing meaningful EDA responses, which could be an issue for short-term applications. Moreover, we observed a high variability in correlation between the forehead and the finger EDA among subjects, which means that for some subjects, the forehead EDA did not produce similar responses to the finger EDA. In cases where finger or palmar sites are not available, we recommend the foot location, because good signals can be obtained from it. The use of the feet may be preferred in the case where hands are required for other daily tasks. For example, divers use their hands for many diving-related tasks, so placing EDA sensors on the feet may be more preferable. We have previously shown that the EDA signal may be used to predict seizures due to oxygen toxicity [20,48]. Hence, a more reliable site other than fingers is needed for applications such as underwater seizure detection.

### Limitations

The first limitation of the study is the lack of sufficient female subjects, which may limit the generalizability of this study. Second, the motion artifacts we considered were limited to a mild level. There could be specific cases where one of the body sites examined might be affected more or less depending on the situation (e.g., diving). We considered walking and weightlifting because these two cases are the most common daily activities. Human subjects typically walk every day and use their hands to perform daily tasks. Moreover, since fingers and feet are two of the best sites to collect EDA signals, we wanted to explore how motion artifacts affected those two locations. For signals with higher motion artifacts, our recently developed convolutional autoencoder-based model may be applied to recover motion artifact-free clean EDA signals [49]. Another limitation of this work is that we did not consider footwear that is placed over the electrodes. The subjects walked barefoot with electrodes, but we obtained good-quality data despite the movements. As shown in Figure 1, subjects can wear socks over the electrodes, which should hold the electrodes in the attached placements. Hence, while socks will result in less movement artifacts, the humidity buildup may lead to some erroneous EDA signals. This is the issue we will examine in future studies.

## 5. Conclusions

This study collected wearable EDA signals from four different measurement sites and compared them in terms of the within-subject correlations with the finger EDA, which is considered the reference. To the best of our knowledge, this is the first study to examine EDA indices from different measurement sites and compare their efficacy to differentiate induced cognitive stress in subjects. Finally, the study compared the robustness of EDAs obtained at different sites against motion artifacts. Based on the results, we recommend the foot be used as an alternative site for EDA data collection, especially when palmar sites or fingers are not available or suitable for placing EDA electrodes. The forehead is a potential alternative site for EDA data collection because of its comparative robustness against motion artifacts, however its low sensitivity to stimuli, prolonged duration of hydration time, and high variability among subjects all suggest that it is not an optimal site; thus, the foot is preferred.

## Figures and Tables

**Figure 1 sensors-22-03177-f001:**
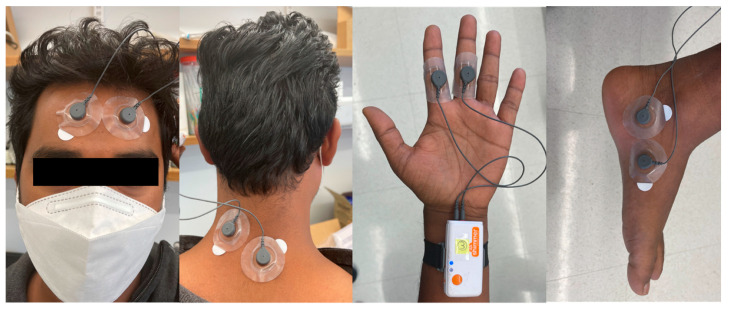
Different locations of electrode placement.

**Figure 2 sensors-22-03177-f002:**
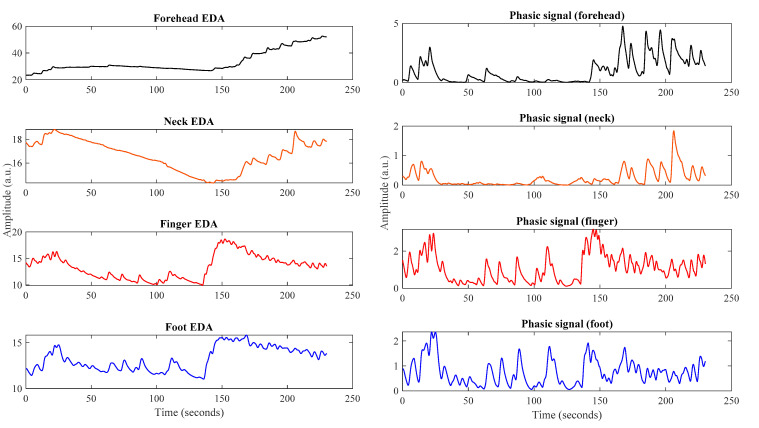
EDA signals collected from different body sites (**left** side), Corresponding phasic signals (**right** side).

**Figure 3 sensors-22-03177-f003:**
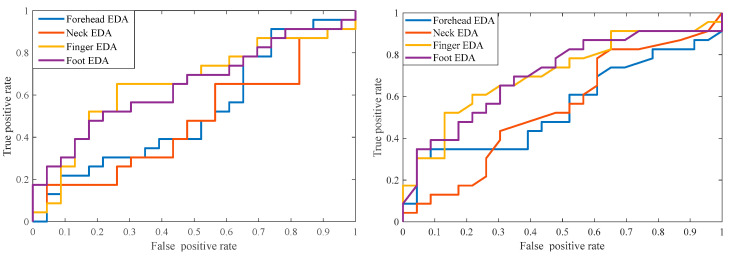
ROC curves for mean of the phasic signal (**left**) and number of SCR (**right**).

**Figure 4 sensors-22-03177-f004:**
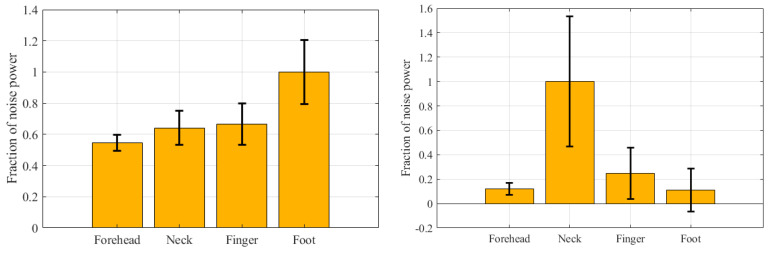
Relative power in the noise band during walking (**left**) and weightlifting (**right**).

**Figure 5 sensors-22-03177-f005:**
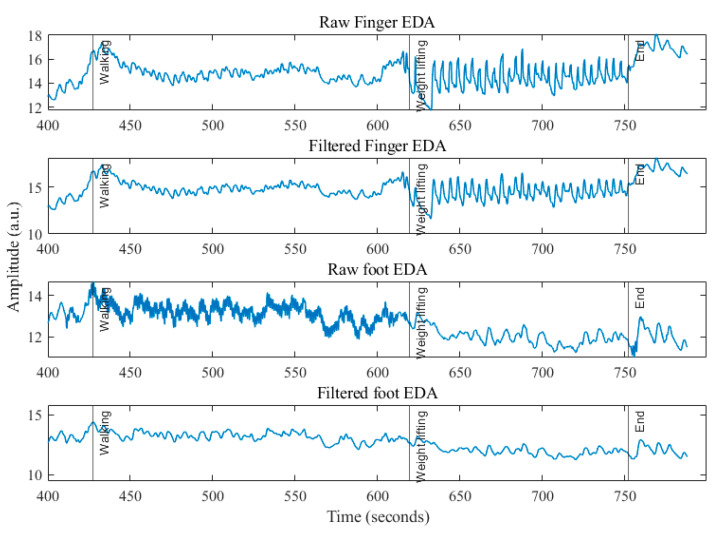
Raw and filtered EDAs (finger and foot) during walking and weightlifting.

**Figure 6 sensors-22-03177-f006:**
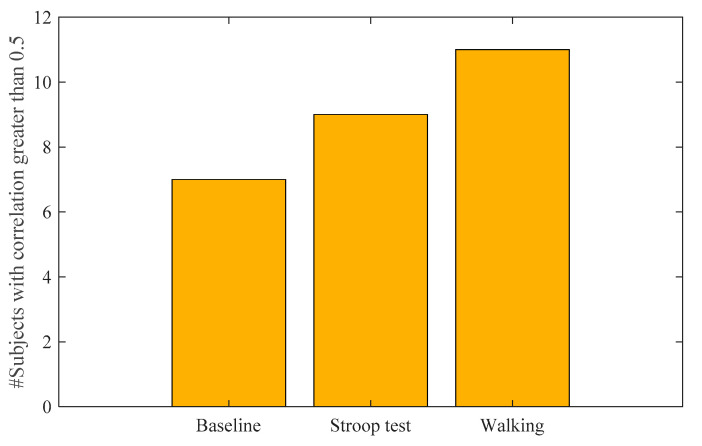
Effect of hydration time on forehead EDA.

**Table 1 sensors-22-03177-t001:** Experimental protocol.

Duration (s)	Activity Description
120	Relax, supine, with eyes closed
120	Perform Stroop test
120	Walk at 3 mph
120	Dumbbell deadlift, one handed; release between each repetition

**Table 2 sensors-22-03177-t002:** Mean and standard deviation of the correlation coefficients between EDA signals from the finger and other body sites.

Location	Raw Data		Phasic		Tonic	
	Mean Pearson r	SD Pearson r	Mean Pearson r	SD Pearson r	Mean Pearson r	SD Pearson r
Forehead	0.3194	0.4510	0.2833	0.2650	0.3077	0.4643
Neck	0.4405	0.4601	0.2530	0.1920	0.4532	0.4200
Feet	0.8466	0.1851	0.7895	0.1118	0.8261	0.1704

**Table 3 sensors-22-03177-t003:** Statistical t-test between the resting and SCWT stages.

Measurement Site	EDA Index	Resting Mean	SCWT Mean	*p*-Value
Forehead	No. of SCR	12.3478	12.3478	0.4896
	Phasic mean	0.2839	0.4378	0.7634
	Phasic variance	0.3040	0.3497	0.9761
	Tonic mean	25.7738	25.6239	0.8692
	Tonic variance	0.9782	1.6313	0.1559
Neck	No. of SCR	5.2609	4.5652	0.5065
	Phasic mean	0.0373	0.0786	0.3799
	Phasic variance	0.0512	0.1030	0.3182
	Tonic mean	4.1619	3.6539	0.1851
	Tonic variance	0.5426	0.4944	0.8788
Finger	No. of SCR	6.0869	11.8696	0.0017 **
	Phasic mean	0.1353	0.2762	0.0016 **
	Phasic variance	0.1362	0.2047	0.0248 *
	Tonic mean	6.5960	6.6051	0.9802
	Tonic variance	1.0982	0.8353	0.1306
Foot	No. of SCR	8.3478	15.0870	0.0006 ***
	Phasic mean	0.1435	0.3562	0.0154 *
	Phasic variance	0.0434	0.1065	0.0091 **
	Tonic mean	5.5115	6.3622	0.0192 *
	Tonic variance	0.5430	0.6312	0.4273

* *p* < 0.05, ** *p* < 0.01, *** *p* < 0.001.

## Data Availability

Not applicable.

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
