# Peer review of "Comparison of Electrodermal Activity from Multiple Body Locations Based on Standard EDA Indices’ Quality and Robustness against Motion Artifact"

_sensors, 2022, doi:10.3390/s22093177_

Round 1
Reviewer 1 Report
The manuscript is focused on the search of body locations suitable for the electrodermal activity data collection. The topic is of interest for medicinal diagnostics and monitoring purposes. The manuscript is well-prepared. The experiment is logical and well-explained. The results are in line with the known facts on the topic. Conclusions are meaningful and supported by experimental data.
There are seeral minor points to be considered prior to acceptance to publication.
- Abbreviation need revision. ANS, ADHD, IRD, have to be removed as far as used just 1-2 times. SCL should be removed and presented in full as "skin conductance level".
- Section 2.2, a short desciption of cvxEDA algorithm can be presented.
- Fig. 4, each column on the plots have to be presented with the error bar.
Author Response
We would like to express our gratitude towards the reviewer for the positive judgment of our work and for the remarks, which helped to improve our paper significantly. We include a point-by-point reply to each of the comments here. We have modified our paper following these comments and we hope that it is now suitable for publication. Please note that all changes to the text are highlighted in yellow.
General Comments:
- The manuscript is focused on the search of body locations suitable for the electrodermal activity data collection. The topic is of interest for medicinal diagnostics and monitoring purposes. The manuscript is well-prepared. The experiment is logical and well-explained. The results are in line with the known facts on the topic. Conclusions are meaningful and supported by experimental data. There are seeral minor points to be considered prior to acceptance to publication.
Thank you for the positive assessment of our work. We have made our best efforts to reorganize the contents of the revised manuscript as you have suggested.
Reviewer#1, Concern # 1:
- Abbreviation need revision. ANS, ADHD, IRD, have to be removed as far as used just 1-2 times. SCL should be removed and presented in full as "skin conductance level".
Thanks for your comment. We removed the abbreviations as the reviewer suggested.
Reviewer#1, Concern # 2:
- Section 2.2, a short desciption of cvxEDA algorithm can be presented.
We appreciate the reviewer’s suggestion and added a paragraph about cvxEDA. We added the following texts.
“We decomposed the EDA signal into phasic and tonic components using the cvxEDA algorithm [40] which is one of the most popular methods for EDA signal decomposition. The cvxEDA models EDA signal as a summation of phasic component, tonic component, and an additive white Gaussian noise term which incorporates the prediction as well as the measurement error. This method then use Bayesian statistics, convex optimization, and sparsity constraint to solve the optimization problem to minimize the prediction errors and obtain the tonic and phasic components.”
Reviewer#1, Concern # 3:
- 4, each column on the plots have to be presented with the error bar.
Thanks for the suggestion. The error bars have been added to the plots.
Reviewer 2 Report
The paper analyzes the Electrodermal Activity (EDA) signals acquired from people performing activities such as walking and lifting a weight, with sensors placed in different body locations (forehead, neck, finger, and foot). The authors compute some EDA parameters, considering the different measurement sites, and compare them in order to explore their efficiency in identifying cognitive stress in subjects, and their robustness against motion artifacts.
The paper is well written and presents an experiment where various EDA signals are logged from different positions in the subject’s body that can be of some interest to the scientific community. However, in my opinion, the final contribution of the manuscript appears to be quite limited to be published as a full paper, and it could be more appropriate for some conference or a short contribution. As a matter of fact, the performance of the system and the results depend on the specific setup considered, and on the kind of activities the subjects are performing. In other words, when considering the particular experiment described in the work, the conclusion about the foot EDA location being the best one seems to be somehow expected, since there is a higher sweat gland density and the particular setup makes motion artifacts easily removable. So, additional data logged from subjects during other movements and performing a wider range of activities, lasting more than two minutes each, should have been included in the paper. This would help in making the work more complete, and would make the conclusions about the favourable use of the foot position more justified in a larger set of applications.
Other issues that should be addressed are as follows. The experimental protocol should be better explained in Section 2.1. Some comments about the duration (two minutes) of each phase should be given. Is there a pause between the different phases? When it is stated that “the Stroop test is performed in the same position”, does it mean that the subjects are supine during the Stroop test as well? In Section 2.2, some choices about the frequency used to process the signals, and the selected EDA parameters, should be better motivated. Please also explain how the number of skin conductance responses is computed. Considering this sentence: “… and the mean and variance from each segment…”, what does “segment” refer to? A brief introduction about the components of EDA (phasic/tonic) may also prove useful for the reader. As for the power computed in the noise band, it seems that all the signal components appearing in the [0.4-0.6] Hz frequency range are considered to be noise or artifacts not related to the EDA signal, why do you make this hypothesis? Please also add some details about how the spectral power is computed. As far as the Results Section is concerned, please explain better how the Pearson correlation is computed: for each subject, are all of the signal samples in the different stages of the experiment considered? The “within-subject” and “between-subjects” terms used in Section 3.1 can be quite confusing: “between-subjects” can mislead the reader to think that the correlation is computed among different subjects. Which normality test has been used? How many data samples are used for the statistical tests? I suppose that each vector being analyzed through the paired t-test/Wilconxon test contains 23 elements, each one corresponding to a subject, but this has to be better clarified in the paper. In addition, how the ROC has been derived should be better described in the paper. Figure 6 and the related explanation about the effect of hydration time need to be clarified in the text.
Some minors:
• in the Introduction, some cited papers appear in regular font and others in bold
• in the Introduction, why are there white spaces among paragraphs?
• row 169: sites[42] -> sites [42]
• row 171: unnecessary white space among adjacent lines
• row 194: Table 3 -> Table 2
• row 231: Fig. 4 Shows -> Fig. 4 shows
• row 268: Fig. 6. shows -> Fig. 6 shows
Author Response
We would like to express our gratitude towards the reviewer for the positive judgment of our work and for the remarks, which helped to improve our paper significantly. We include a point-by-point reply to each of the comments here. We have modified our paper following these comments and we hope that it is now suitable for publication. Please note that all changes to the text are highlighted in yellow.
General Comments:
The paper analyzes the Electrodermal Activity (EDA) signals acquired from people performing activities such as walking and lifting a weight, with sensors placed in different body locations (forehead, neck, finger, and foot). The authors compute some EDA parameters, considering the different measurement sites, and compare them in order to explore their efficiency in identifying cognitive stress in subjects, and their robustness against motion artifacts.
Thank you for the positive assessment of our work. We have made our best efforts to reorganize the contents of the revised manuscript as you have suggested.
Reviewer#2, Concern # 1:
- The paper is well written and presents an experiment where various EDA signals are logged from different positions in the subject’s body that can be of some interest to the scientific community. However, in my opinion, the final contribution of the manuscript appears to be quite limited to be published as a full paper, and it could be more appropriate for some conference or a short contribution. As a matter of fact, the performance of the system and the results depend on the specific setup considered, and on the kind of activities the subjects are performing. In other words, when considering the particular experiment described in the work, the conclusion about the foot EDA location being the best one seems to be somehow expected, since there is a higher sweat gland density and the particular setup makes motion artifacts easily removable. So, additional data logged from subjects during other movements and performing a wider range of activities, lasting more than two minutes each, should have been included in the paper. This would help in making the work more complete, and would make the conclusions about the favourable use of the foot position more justified in a larger set of applications.
Thank you for the comments. Given that this is a non-methodological paper, the contribution of the paper might be less evident. However, we would like to highlight the following contributions of this work. Note that for some practical cases, EDA signals from fingers are not possible as the sensors on the fingers may get in the way of performing one’s job duties. Hence, the purpose of this work was to find alternative sites other than fingers for EDA measurements. To this end, there have been two prior studies which examined the best location for EDA sensor placement other than fingers. The results have been conflicting for the signal quality obtained from other locations. This may be due to analysis involving calculation of simple correlations between finger and other body locations of EDA signals. Given recent advances in the way EDA signals are quantitatively analyzed involving several sophisticated methods, the aim was to use these approaches to determine other body locations to obtain good EDA signals other than fingers. In addition, we examined other alternative body locations for EDA measurements considering how those sites may fare against motion artifacts. These aims were not examined in prior studies from non-engineering journals and we believe that these are important matters to address.
Regarding the experimental setup we agree with the reviewer that our motion artifact conditions are limited (mentioned in the limitations) and it would have been optimal to include MAs from a wide range of activities including ambulatory monitoring. However, it should be stressed that we are using multiple sensors, and wearing them for ambulatory monitoring is challenging and inconvenient. Therefore, we considered two special MA conditions that are very common and certainly affect the signal quality of the finger and foot EDA. For recovering MA-corrupted EDA data, our recently developed convolutional autoencoder model can be applied which has shown promising performance [49]. Moreover, we believe there will be cases where EDA data are severely corrupted. In such case, severely-corrupted EDA data portion can be discarded automatically using recently-developed machine learning method for automatic MA detection [45]. Nevertheless, we really appreciate the reviewer’s idea of collecting EDA data from multiple locations in wider MA scenarios which we will definitely pursue in the future.
Reviewer#2, Concern # 2:
- Other issues that should be addressed are as follows. The experimental protocol should be better explained in Section 2.1. Some comments about the duration (two minutes) of each phase should be given. Is there a pause between the different phases? When it is stated that “the Stroop test is performed in the same position”, does it mean that the subjects are supine during the Stroop test as well?
We have updated the description of the experimental protocol to provide more details. We added the following texts.
“Part I was designed to compare the EDA collected with no movement as subjects were resting in the supine position for two minutes followed by the Stroop test in the same position (supine position). The experimental phases were performed one after another with no pause in between. For each phase we considered 2 minutes which is sufficiently enough to contain many SCRs.”
- In Section 2.2, some choices about the frequency used to process the signals, and the selected EDA parameters, should be better motivated. Please also explain how the number of skin conductance responses is computed. Considering this sentence: “… and the mean and variance from each segment…”, what does “segment” refer to? A brief introduction about the components of EDA (phasic/tonic) may also prove useful for the reader.
Thanks for your thoughtful suggestions. We provided some rationale for the choice of the desired frequencies and updated the data processing subsection to clarify the confusion regarding ‘segment’. We also included a brief definition of phasic and tonic components of EDA and provided appropriate references. The updated texts are as below.
“The EDA data were resampled at 8 Hz from 100 Hz. We then used a Butterworth lowpass filter of order six and cutoff frequency of 0.6 Hz (which is well above the frequency range of EDA dynamics) [39]. Since the autonomous nervous activities are typically captured within the frequency range () [40,41], the choice of cutoff frequency of 0.6 Hz was a relatively conservative approach to remove redundancy in the data while keeping the essential information. We decomposed the EDA signal into phasic and tonic components using the cvxEDA algorithm [42] which is one of the popular methods for EDA signal decomposition. The cvxEDA models EDA signal as a summation of phasic and tonic components, and an additive white Gaussian noise term, which combines the prediction as well as the measurement error. This method then uses Bayesian statistics, convex optimization, and sparsity constraint to solve the optimization problem to minimize the prediction errors to estimate the tonic and phasic components. Phasic and tonic components are two most salient characteristics of an EDA signal, where the phasic components (SCRs) represents the rapid and smooth transient events present in the EDA signals, and the tonic component represents the overall conductance as a measure related to the slow shifts of the EDA [43].
We then used the information noted during the experiment to segment the data into different stages, where each segment corresponds to a particular phase of the experiment (e.g. baseline, Stroop test, walking, and weightlifting). We computed traditional EDA indices such as the number of skin conductance responses, mean and variance of skin conductance level, and the mean and variance of phasic signals from each segment of the four EDA locations [43]. Since EDA signals collected from different locations may have different signal amplitudes of the SCRs, we hence used, 0.05 of the maximum peak amplitude of the respective phasic signal as the threshold to compute the number of skin conductance response.”
- As for the power computed in the noise band, it seems that all the signal components appearing in the [0.4-0.6] Hz frequency range are considered to be noise or artifacts not related to the EDA signal, why do you make this hypothesis? Please also add some details about how the spectral power is computed.
The reviewer raises an interesting issue. As mentioned earlier the most of the ANS activity are obtained within the frequency range () which is why we considered the frequency contents (>0.4 Hz) as redundant or the noise band. We provided more justification in the revised manuscript with appropriate references. We also provided details on how spectral power is calculated. We updated the texts as follows.
“To compare the data quality among different body locations, we first computed the power spectrum of the EDA signal using Welch periodogram with 50% data overlap, and then defined a noise band (frequency > 0.4 Hz). The reason we chose 0.4 Hz as the cutoff frequency is that most of the ANS activity (sympathetic and parasympathetic) are captured within the frequency range 0.04 to 0.4 Hz [39,40]. Therefore, we can attribute the spectral power in the frequency > 0.4 Hz to either noise or dynamics not related to the EDA signal.”
- As far as the Results Section is concerned, please explain better how the Pearson correlation is computed: for each subject, are all of the signal samples in the different stages of the experiment considered?
Thanks for the comment. We added the following texts to clarify how we computed the Pearson correlations.
“We computed the Pearson correlation between the finger EDA and EDAs from other body sites during the “no movement” phases (i.e., baseline and SCWT). Since for the first two phases of the protocol there is no significant motion artifact, for accurate correlation calculation, we used continuous 4 minutes data (baseline and SCWT) for calculating Pearson correlation.”
- The “within-subject” and “between-subjects” terms used in Section 3.1 can be quite confusing: “between-subjects” can mislead the reader to think that the correlation is computed among different subjects.
Thanks for pointing that out. We have explained each term in the brackets to avoid any confusion. The corrected sentences are:
“Table 2 shows within-subject correlations (correlation is computed among the EDA channels of the same subject) between the finger EDA and the EDAs from the alternate sites”
“We also observed a larger between-subjects variability (higher variability across the subjects: evident from higher standard deviation value), which suggests that even though for some subjects the forehead and neck EDA were highly correlated with the finger EDA, for others, this was not the case.”
- Which normality test has been used? How many data samples are used for the statistical tests? I suppose that each vector being analyzed through the paired t-test/Wilconxon test contains 23 elements, each one corresponding to a subject, but this has to be better clarified in the paper.
Thanks for the comment. We included the details about the statistical tests in the revised manuscript. We updated the texts as below.
“We first performed one-sample Kolmogorov-Smirnov test [47] on the data, and if normality is found, we performed the statistical t-test between baseline and SCWT stage for each EDA index. Otherwise we used Wilcoxon rank sum test [44]. All the statistical test was performed across the subjects over 23 samples. We used the 0.05 level of significance for rejecting the null hypothesis. Table 3 shows the summary results for each measurement site. “
- In addition, how the ROC has been derived should be better described in the paper. Figure 6 and the related explanation about the effect of hydration time need to be clarified in the text.
Thank you for your comment. We added details on how ROC curve is computed and provided more explanations in figure 6. We added the following texts.
“We first computed true positive rate and false positive rate for each feature at a variety of thresholds. ROC is curve is then obtained by plotting true positive rate against false positive rate. In our case, the positive class refers to cognitive stress stage and the negative class represents the baseline stage.”
Regarding the hydration time we discussed that forehead need more hydration time for producing accurate SCRs. We modified the paragraph to provide clearer description. The revised paragraph is as follows.
“The effect of hydration time on the forehead EDA can be seen when we examine the number of subjects that showed correlation higher than 0.5 between the finger and the forehead or the neck in different stages starting from the baseline. Fig. 6 shows the progression of the number of subjects showing correlation greater than 0.5 in different stages. As shown, as time progressed, more subjects had higher correlations between the forehead and finger EDA, which indicates that with sufficient hydration the forehead may become more responsive and provide more accurate SCRs.”
Reviewer#2, Concern # 3:
- Some minors:
in the Introduction, some cited papers appear in regular font and others in bold
• in the Introduction, why are there white spaces among paragraphs?
• row 169: sites[42] -> sites [42]
• row 171: unnecessary white space among adjacent lines
• row 194: Table 3 -> Table 2
• row 231: Fig. 4 Shows -> Fig. 4 shows
• row 268: Fig. 6. shows -> Fig. 6 shows
We thank the reviewer for finding the typos or errors. They are corrected according to the reviewer’s suggestions.
Reviewer 3 Report
Interesting paper.
First thing I notice is the poor formatting. The paper should be fully justified, as in the template. Likewise, there should not be a blank line above paragraphs. Sometimes tables are missing. A careful review of how the authors have not adhered to the template should be done.
Table 1 is presented before it is discussed in the main text – needs to move.
Equation 1 seems to be presented in an incorrect font size.
The reference list is incorrectly presented.
After reading the paper, I can see the point of the paper, I have some comments that do not seem to be addressed in the paper:
- While there is a high correlation between finger EDA and foot EDA, OK, but what do the authors intend to do about it? What applicant do they target their work for? What system do they consider that would do EDA on the heel? A research article needs t have a point – but this paper finds something out and does not do anything with the result.
- Foot EDA is difficult due to socks, shoes and the temperature that gets held by the footwear. Please comment on if your results are valid when wearing shoes, or is it just bear feet you consider? If it is just bear feet, what applicant do you consider that would be valid for bear feet?
Author Response
We would like to express our gratitude towards the reviewer for the positive judgment of our work and for the remarks, which helped to improve our paper significantly. We include a point-by-point reply to each of the comments here. We have modified our paper following these comments and we hope that it is now suitable for publication. Please note that all changes to the text are highlighted in yellow.
General Comments:
Interesting paper.
Thank you for the positive assessment of our work. We have made our best efforts to reorganize the contents of the revised manuscript as you have suggested.
Reviewer#3, Concern # 1:
- First thing I notice is the poor formatting. The paper should be fully justified, as in the template. Likewise, there should not be a blank line above paragraphs. Sometimes tables are missing. A careful review of how the authors have not adhered to the template should be done.
We appreciate your comment. We did our best to format the paper properly in the revised manuscript. Moreover, we believe once the paper is accepted it will be properly formatted by the journal before the publications.
Reviewer#3, Concern # 2:
- Table 1 is presented before it is discussed in the main text – needs to move.
We have moved the Table 1 so that it does not appear before it is described.
Reviewer#3, Concern # 3:
- Equation 1 seems to be presented in an incorrect font size.
We thank the reviewer for finding the error. We used correct font in the revised manuscript.
Reviewer#3, Concern # 4:
- The reference list is incorrectly presented.
We have reformatted the paper and the references. We hope reference list is according to the correct format now.
Reviewer#3, Concern # 5:
- After reading the paper, I can see the point of the paper, I have some comments that do not seem to be addressed in the paper:
- While there is a high correlation between finger EDA and foot EDA, OK, but what do the authors intend to do about it? What applicant do they target their work for? What system do they consider that would do EDA on the heel? A research article needs t have a point – but this paper finds something out and does not do anything with the result.
- Foot EDA is difficult due to socks, shoes and the temperature that gets held by the footwear. Please comment on if your results are valid when wearing shoes, or is it just bear feet you consider? If it is just bear feet, what applicant do you consider that would be valid for bear feet?
The reviewer raises some important points. We added the following texts in the discussion to provided target applications of the findings of the work.
““In cases where finger or palmar sites are not available, we recommend the foot EDA because good signals can be obtained from it. The use of the feet may be preferred in the case where hands are required for other daily tasks. For example, divers use their hands for many diving related tasks so placing EDA sensors on the feet may be more preferable. We have previously shown that EDA signal may be used to predict seizures due to oxygen toxicity [20][48]. Hence, a more reliable site other than fingers is needed for application such as underwater seizure detection.”
Regarding the footwear issue, we performed the experiment barefoot. However, we believe wearing socks or footwear should not change the EDA data quality significantly, since socks can be worn on top of the electrodes which should lead to better adhering of the electrodes to the skin. Nevertheless, we mentioned this issue as a limitation of the study. We added the following texts.
““Another limitation of this work is that we did not consider a footwear that is placed over the electrodes. The subjects walked barefoot with electrodes, but we obtained good quality data despite movements. As shown in Fig. 1, subject can wear socks over the electrodes, which should hold the electrodes in the attached placements. Hence, while socks will result in less movement artifact, the humidity buildup may lead to some erroneous EDA signals. This is the issue we will examine in future studies.”
Round 2
Reviewer 2 Report
The paper has been modified according to some of my previous comments. In particular, in the new version of the manuscript, the authors provided additional details about the experimental setup, the selected EDA parameters and the frequency range used to process the EDA signals. They also added some clarifications about the statistical tests performed on the data. I recognize the efforts made by the authors in improving the quality of the paper.
However, as already noted in my previous review, the reported results and the conclusions refer to a specific and somewhat unrealistic setup, and depend on the type of activities the subjects are doing. I still think that additional data acquired from subjects performing other activities and movements should be analyzed.
Author Response
General Comments:
- The paper has been modified according to some of my previous comments. In particular, in the new version of the manuscript, the authors provided additional details about the experimental setup, the selected EDA parameters and the frequency range used to process the EDA signals. They also added some clarifications about the statistical tests performed on the data. I recognize the efforts made by the authors in improving the quality of the paper.
Thank you for acknowledging our effort to address the reviewer’s comments.
Reviewer#1, Concern # 1:
- However, as already noted in my previous review, the reported results and the conclusions refer to a specific and somewhat unrealistic setup and depend on the type of activities the subjects are doing. I still think that additional data acquired from subjects performing other activities and movements should be analyzed.
Please note that the journal allowed only 10 days for the revised paper to be submitted. Given the short time limit, we collected 20 minutes of ambulatory EDA data (sitting, walking, typing, etc.) from 2 subjects using the same body locations and electrode placement settings as described in the manuscript. The figures below show plots of two subjects’ 20 min EDA data from multiple body locations.
As shown in these figures, signals obtained from the finger and the foot are quite similar as reported in the manuscript for both subjects. We further quantitatively analyzed the effect of motion artifact on EDA data from the foot and finger. We did not analyze EDA data from the forehead and neck since they were not as good the other two locations. We performed motion artifact detection on the foot and finger EDA data, and computed the percentage of samples detected as MAs. The percentage of samples detected as MAs for the finger and the foot EDA were 5.37% and 6.57%, respectively, which indicate that in terms of usability in an ambulatory setting, the foot and the finger EDA data quality are nearly identical. While these results are on a limited number of subjects, they are consistent with those already reported in the manuscripts.
|
||||
|
||||

Reviewer 3 Report
The reviewers have reflected on my comments and incorporated most of them.
It is disappointing that the authors have not addressed the poor formatting comment. I don’t understand why they would not want to present a correctly presented paper.
Author Response
- It is disappointing that the authors have not addressed the poor formatting comment. I don’t understand why they would not want to present a correctly presented paper.
We apologize that we did not address the formatting issue as you’ve recommended. We have now properly formatted the paper. Thank you.